# Regulatory T Cells Induce Metastasis by Activating Tgf-Β and Enhancing the Epithelial–Mesenchymal Transition

**DOI:** 10.3390/cells8111387

**Published:** 2019-11-04

**Authors:** Eonju Oh, JinWoo Hong, Chae-Ok Yun

**Affiliations:** 1Department of Bioengineering, College of Engineering, Hanyang University, 222 Wangsimni-ro, Seongdong-gu, Seoul 04763, Korea; djswn1111@hanyang.ac.kr (E.O.); jhong803@gmail.com (J.W.H.); 2Institute of Nano Science and Technology (INST), Hanyang University, 222 Wangsimni-ro, Seongdong-gu, Seoul 04763, Korea

**Keywords:** melanoma, cancer metastasis, epithelial–mesenchymal transition, TGF-β, regulatory T cells

## Abstract

Malignant melanoma is the most aggressive form of skin cancer; a substantial percentage of patients present with distant metastases. However, the mechanism of metastasis is not well understood. Here, we demonstrate that the administration of exogenous regulatory T cells (Tregs) into melanoma tumor-bearing mice results in a significant increase in lung metastasis. An increase in the invasive and metastatic phenotype of melanoma was mediated by cell-to-cell contact between melanoma cells and Tregs, which elevated the expression level of transforming growth factor-β (TGF-β) and the subsequent induction of the epithelial-to-mesenchymal transition (EMT). B16-BL6 melanoma tumors co-cultured with Tregs showed a larger population of migrating cells compared to B16-BL6 tumors cultured without Tregs. Additionally, the injection of exogenous Tregs into B16-BL6 melanoma tumors led to the recruitment and infiltration of endogenous Tregs into tumor tissues, thus increasing the overall Treg percentage in the tumor infiltrating lymphocyte population. Collectively, our findings propose novel mechanisms in which exogenous Treg-dependent upregulation of TGF-β and mesenchymal markers is important for augmenting the migration capacity and invasiveness of melanoma, thereby contributing to the metastasis.

## 1. Introduction

Skin cancer is categorized into 3 types, which are named on the basis of the cell from which they arise or their clinical behavior: basal cell carcinoma, squamous cell carcinoma, and malignant melanoma [1]. Malignant melanoma is the most aggressive form of skin cancer, with an incidence rate that is continuously increasing worldwide. In general, mortality in patients has remained stable or is slightly decreasing due to early diagnosis as a result of prevention efforts, such as public education and increased surveillance [2,3,4]. Although the earliest stages of melanoma can be cured by surgical resection, metastatic melanoma in the advanced stage of the disease is notoriously difficult to treat [2,5,6,7]. Melanoma patients with metastases in 3 or more regions have a 1-year survival rate of less than 5% [5,8]. Thus, successful disease management of advanced melanomas requires innovative interventions that can efficiently treat metastatic disease. 

Various molecular pathways that underlie the multistage process of metastasis formation have been characterized [9,10]. The complex, sequential steps leading to metastasis involve a crucial process called the epithelial–mesenchymal transition (EMT), which endows cancer cells with migratory and invasive capabilities associated with mesenchymal phenotype [11,12,13]. EMT is induced by a variety of growth factors, such as transforming growth factor-β (TGF-β) hepatocyte growth factor, insulin-like growth factor, and epidermal growth factor [10,14]. In particular, TGF-β plays an important part in cancer cell migration, invasion, and metastatic dissemination [15,16].

Numerous studies have documented that TGF-β is a key regulator promoting invasiveness and the metastatic potential of melanoma [17,18,19,20]. In addition, TGF-β has direct and indirect suppressive activity against cluster of differentiation (CD)4^+^ T cells, CD8^+^ T cells, and natural killer cells, thus promoting immunosuppression in the tumor microenvironment. Another mechanism of TGF-β-induced immunosuppression occurs through the activation of regulatory T cells (Tregs), which serve as functional suppressors of the antitumor immune response [17]. Interestingly, melanoma expresses TGF-β-induced forkhead box P3 transcription factor (Foxp3), which functions as an intracellular molecular marker of Tregs [21]. These two aspects of TGF-β function are connected as cancer metastasis is accelerated through immunosuppression [22].

Although the accumulation of Tregs within the tumor microenvironment has been associated with a poor prognosis [23,24,25,26,27], the role of Tregs in metastasis and EMT of melanoma cells is poorly understood. To acquire a better understanding of Treg-induced metastasis in melanoma, we examined in vitro and in vivo effects of Tregs on the EMT, invasion, migration, and metastasis of melanoma, ultimately showing that Tregs induce metastasis by promoting TGF-β expression and EMT.

## 2. Materials and Methods

### 2.1. Cell Lines and Culture Conditions

Murine melanoma cell lines (B16-F10 and B16-BL6) were purchased from the American Type Culture Collection (ATCC, Manassas, VA, USA). B16-F10 cells were cultured in Dulbecco’s modified high-glucose Eagle’s medium (DMEM, PAN Biotech, Dorset, UK) and B16-BL6 cells in modified Eagle’s medium (Life Technologies, Gaithersburg, MD, USA). All media were supplemented with 10% fetal bovine serum (FBS; PAN Biotech) and 100 IU/mL penicillin-streptomycin (PAN Biotech). Additionally, the medium was supplemented with 1 mmol/L modified Eagle’s medium vitamin solution (Gibco BRL, Grand Island, NY, USA) to culture B16-BL6 cells. All cell lines were maintained at 37 °C in a humidified atmosphere containing 5% CO_2._

### 2.2. Mice

Five-week-old C57BL/6 mice were obtained from Orient Bio Inc. (Seongnam, Korea). Green fluorescent protein (GFP) transgenic mice were purchased from Jackson Laboratories (Bar Harbor, ME). During the experiments, mice were maintained in a laminar air-flow cabinet with specific pathogen-free conditions. All facilities were approved by the Association for Assessment and Accreditation of Laboratory Animal Care. All animal studies were conducted according to the institutional guidelines established by the Hanyang University Institutional Animal Care and Use Committee.

### 2.3. Isolation, Purification, and Differentiation of Regulatory T Cells (Tregs) from Draining Lymph Nodes

100-mm dishes were coated with anti-CD3 antibody (Ab) (cat. no. 553057, clone 145-2C11, BD Pharmingen, San Diego, CA, USA) for 24 h and washed before use. A single cell suspension was prepared from the draining lymph node (DLN) of GFP transgenic mice (Appendix A). These DLN cells (DLNC), which stably express GFP, were cultured in RPMI-1640 (PAN Biotech) supplemented with 10% FBS (PAN Biotech) in the presence of 1000 U/mL recombinant interleukin-2 (rIL-2) (R&D system, Minneapolis, MN), 2 μg/μL rCD28 (BD Pharmingen), and rTGF-β (R&D system) to promote differentiation into Tregs. At 7 days of culture *ex vivo* in supplemented medium, CD4^+^ T cells were isolated using MagCellect Mouse CD4^+^ T cell isolation kit (R&D system) according to the manufacturer’s protocol. After magnetic cell separation, these cells were further sorted by cell sorting using a FACSAria^TM^ III sorter (BD Biosciences, San Jose, CA, USA). 

### 2.4. Migration and Invasion Assays

Migration and invasion assays were performed as described previously [28,29]. The lower surfaces of 6.5 mm polycarbonate filters (8 μm pore size; Corning Costar, Cambridge, MA, USA) were coated by immersion in 0.1% gelatin. B16-F10 cells, which were placed on the filter membrane in the top portion of a transwell chamber, were co-cultured with DLNC or Tregs at various co-culture ratios. Normal culture medium (DMEM with 10% FBS) was placed in the lower part of the transwell chambers. Cultures were incubated at 37 °C for 48 h, fixed in methanol, and stained with hematoxylin and eosin (H & E). To assess the migration of dissociated tumor cells, B16-BL6 tumors were injected intratumorally 3 times every other day with Tregs (2 × 10^7^ cells). Alternatively, B16-BL6 cells were co-cultured with Tregs for 72 h at co-culture ratios of 1:10. Co-cultured cells were then washed multiple times with phosphate-buffered saline (PBS) to remove inadherent Tregs from the culture prior to trypsinization. Subsequently, 5 × 10^5^ cells were counted then injected subcutaneously into the right abdomen of 6- to 7-week-old male C57BL/6 mice to establish a tumor. B16-BL6 tumors directly injected with Tregs were collected at day 5 following the final Tregs injection, whereas Tregs-co-culture-induced B16-BL6 tumors were harvested at 15 days after the subcutaneous inoculation of tumor cells. Dissociated tumors were prepared as previously described [30], whereas migration assays were performed as described above. Matrigel invasion assays were performed using transwell invasion chambers coated with Matrigel (BD Biosciences). The experiment was performed as described for the cell migration assay. After 72 h, non-invading cells were removed, and the invading cells on the lower surface of the filter were fixed and stained. The membranes were mounted on glass slides, and migrated cells were counted at ×200 magnification. 

### 2.5. Quantification of Transforming Growth Factor-β (TGF-β) Expression

B16-F10 cells were plated onto 6-well plates at a density of 1 × 10^5^ cells per well, and then co-cultured with DLNC or Tregs at various co-culture ratios while cell-to-cell contact was allowed. Alternatively, B16-F10 cells seeded as described above were co-incubated with DLNC or Tregs while cell-to-cell contact was prohibited using a 24-well transwell chamber. B16-F10 cells were plated onto 24-well plates in lower chamber at a density of 2 × 10^4^ cells per well and DLNC or Tregs were placed in upper chamber at various co-culture ratios. After 72 h of incubation, supernatants in lower chambers were collected. TGF-β expression was determined by using a TGF-β enzyme-linked immunosorbent assay (ELISA) kit (R&D Systems) according to the manufacturer’s protocol. 

### 2.6. Western Blot Analysis

B16-F10 cells were co-cultured with DLNC or Tregs at various co-culture ratios for 72 h. Western blotting was performed as described previously [31]. Blocked membranes were incubated with primary Abs against Foxp3 (cat. no. ab54501, abcam, Cambridge, MA, USA), TGF-β (cat. no. ab9758, abcam), Smad2/3 (cat. no. 8685, clone D7G7, Cell signaling technology, Beverly, MA), β-catenin (cat. no. 9587, Cell signaling technology), α-SMA (alpha-smooth muscle actin; cat. no. ab5694, abcam), vimentin (cat. no. 3932, clone R28, Cell signaling technology), or MMP9 (Matrix metalloproteinase 9; cat. no. ab137867, clone EP1255Y, abcam) overnight at 4 °C. The blots were incubated with the following secondary Abs conjugated to horseradish peroxidase: goat anti-rabbit IgG (cat. no. 7074, Cell signaling technology), goat anti-mouse IgG (cat. no. 7076, Cell signaling technology), or mouse anti-goat IgG (cat. no. 14-13-06, KPL/SeraCare, Gaithersburg, MD, USA) and developed using enhanced chemiluminescence (Amersham Pharmacia Biotech, Uppsala, Sweden). Protein expression was semi-quantitatively analyzed using ImageJ software (version 1.50b; U.S. National Institutes of Health, Bethesda, MD, USA). 

### 2.7. B16-BL6 Spontaneous Lung Metastasis Model 

A spontaneous metastasis model was used to examine the effect of Tregs on tumor metastasis. B16-BL6 metastasis model was established by a method similar to those described in our previous reports [32,33]. Briefly, 1 × 10^5^ B16-BL6 cells were injected subcutaneously into the right hind foot pad. When tumors reached a volume of 80 mm^3^, mice were randomly assigned to 2 different groups and intratumorally injected with PBS or 1 × 10^6^ Tregs (three mice per group). The first day of injection was designated as day 1; 2 additional doses were administered on days 3 and 5. On day 10, the primary tumors were surgically excised by amputating the right hind leg below the knee under anesthesia. On day 25 after primary tumor removal, the total lung tissue weight, combining both normal and cancerous tissues, was assessed. 

### 2.8. Immunohistochemical Analysis

B16-BL6 cells were co-cultured with Tregs for 72 h, and then 5 × 10^5^ cells were injected subcutaneously into the right abdomen of 6- to 7-week-old male C57BL/6 mice. Tumor tissues were harvested from mice at 15 days after cancer cell injection and fixed in 10% formalin, processed for paraffin embedding, and cut into 5 μm-thick sections. Tumor sections were immunostained with rabbit anti-mouse TGF-β, rabbit anti-mouse α-SMA, rabbit anti-mouse β-catenin, rabbit anti-snail/slug, or rabbit anti-vimentin purchased from abcam. After incubating with primary Ab at 4 °C overnight, sections were incubated with Alexa Fluor 488-labeled goat anti-rabbit IgG (cat. no. A11008, Invitrogen, Carlsbad, CA) or Alexa Fluor 568-labeled goat anti-rabbit IgG (cat. no. A11001, Invitrogen) at room temperature for 1 h. For counterstaining, the samples were incubated with 4,6-diamidino-2-phenylindole (Sigma, St. Louis, MO, USA). Slides were viewed under a confocal laser-scanning microscope (LSM510, Carl Zeiss MicroImaging, Thornwood, NY, USA). 

### 2.9. Fluorescence-Activated Cell-Sorting Analysis

To assess the Treg population by flow cytometry, the B16-BL6 tumors were harvested at 5 days following the final Tregs injection. Dissociated tumors were prepared as previously described [30]. Cells were stained as previously described [34]. Briefly, the cells were stained with peridinin chlorophyll protein-CY5.5-conjugated anti-CD4 Ab (cat. no. 550954, clone RM4-5, BD Biosciences), phycoerythrin-conjugated anti-CD25 Ab (cat. no. 12-0251-81, clone PC61.5, eBioscience, San Diego, CA, USA), and allophycocyanin-conjugated anti-Foxp3 Ab (cat. no. 17-5773-82, clone FJK-16s, eBioscience). Samples were analysed using a BD Biosciences BD FACScanto II flow cytometry analyser and FACSDiva software (BD Biosciences).

### 2.10. Statistical Analysis

Data were expressed as mean ± standard deviation (SD). Statistical significance was determined by Independent-samples test (*t*-test for equality of means and Levene’s test for equality of variances) (SPSS 13.0 software; SPSS, Chicago, IL, USA). Data with *P* values less than 0.05 were considered statistically significant.

## 3. Results

### 3.1. Effect of Tregs on Melanoma Migration and Invasion In Vitro

Migratory properties of cancer cells are crucial for the spreading of metastatic tumor cells [35]. To investigate whether Tregs enhance the migration and invasion of melanoma, B16-F10 cells were co-cultured with DLNC or Tregs at various co-culture ratios. Tregs were obtained by culturing DLN-derived CD4^+^ T cells with IL-2, CD28, and TGF-β. B16-F10 cells cultured in the absence of other cell types were used as a negative control. After 72 h of solitary culturing or co-culturing, migration and invasion assays were carried out by utilizing a transwell plate either without or pre-coated with Matrigel, respectively. As shown in Figure 1A, B16-F10 cells co-cultured with DLNC or Tregs showed a greater migration capacity, which increased as the quantity of co-cultured DLNC or Tregs increased, compared with non-co-cultured cancer cells (*p* < 0.01 or *p* < 0.001 for DLNC and Tregs). Importantly, Treg-co-cultured B16-F10 cells showed greater migration than DLNC-co-cultured B16-F10 cells (*p* < 0.01), suggesting that Tregs are one of the most crucial subsets of cells within the DLN for promoting the migration of melanoma cells. Similarly, Tregs also induced a 2.1- or 1.4-fold increase in cancer cell invasion into Matrigel compared with DLNC at co-culture ratios of 1:10 or 1:20, respectively (Figure 1B; *p* < 0.01), showing that both the migration capacity and the invasiveness of melanoma cells are elevated by interaction with Tregs. 

### 3.2. Increased Expression of TGF-β in Melanoma Cells Following Co-Culture with Tregs

The TGF-β pathway regulates cancer progression including cancer metastasis [36]. Because increasing the Treg population appears to enhance cancer cell migration and invasion, we tested whether these effects of Tregs were caused by the induction of TGF-β expression in melanoma cells. As shown in Figure 2A, Treg-co-cultured B16-F10 cells exhibited Treg density-dependent up-regulation of TGF-β expression compared with non-co-cultured cells (*p* < 0.01 or *p* < 0.001). Moreover, Treg-co-cultured B16-F10 cells showed significantly increased TGF-β expression compared to B16-F10 cells co-cultured with DLNC (*p* < 0.01). Of note, the induction of TGF-β required physical contact between Tregs and melanoma cells, as this effect was not observed when these cells were separated in a transwell chamber (Figure 2B). These results suggest that a direct melanoma-Treg interaction is essential for the increased TGF-β production by melanoma cells. 

### 3.3. Increased Expression of Foxp3, Smad2/3, EMT-Related Markers, and MMP9 in Melanoma Cells Following Co-Culture with Tregs 

Foxp3, a key regulator of immune suppression, is expressed by both melanoma cells and Tregs [21]. Therefore, we examined whether Treg-co-cultured B16-F10 cells, which produced markedly higher amounts of TGF-β than DLNC-co-cultured B16-F10 cells, would also show increased levels of Foxp3 expression. As shown in Figure 3A, co-culturing of B16-F10 cells with Tregs as compared with DLNC resulted in greater expression of Foxp3. In addition, the expression level of intracellular signaling components of the TGF-β superfamily (known as Smad2 and Smad3) was increased after exposure of B16-F10 cells to Tregs. These results show that B16-F10 cells produce Foxp3, Smad2, and Smad3 when in contact with Tregs, thus promoting migration and invasion of melanoma cells. 

EMT has been shown to be crucial for cancer progression and metastasis [14]. Thus, we examined whether Treg-induced invasion and migration were caused by elevation of EMT-related markers in B16-F10 cells. We observed increased expression of mesenchymal markers, such as β-catenin, α-SMA, and vimentin, when the cancer cells were co-cultured with Tregs (Figure 3B), indicating that Tregs promote EMT-related events in cancer cells. 

MMP9 plays an important role in promoting migration and invasion of cancer cells [37,38]. Thus, we investigated the effect of Tregs on the expression of MMP9, which induces angiogenesis, tumor growth, and metastasis [39,40,41]. As shown in Figure 3C, pro- and active-MMP9 expression levels were increased in Treg-co-cultured B16-F10 cells, whereas DLNC-co-cultured cells showed slightly increased MMP9 expression compared with non-co-cultured cells. Taken together, these findings suggest that Tregs are a key inducer of multiple cancer metastasis-related factors in melanoma cells.

### 3.4. Immunohistochemical Assessment of Treg Co-Cultured B16-Bl6 Derived Tumor Tissue

To better understand the mechanism behind the Treg-induced increase of cancer metastasis, immunohistochemical analysis was performed on tumors derived from melanoma cells injected into the mouse abdomen. Tumors were harvested at 15 days after subcutaneous injection of Treg-co-cultured B16-BL6 cells. We found that TGF-β expression was significantly increased in tumors derived from Treg co-cultured B16-BL6 cells compared with non co-cultured B16-BL6 cells (Figure 4A, *p* < 0.001), supporting the notion that Tregs are one of the main inducers of TGF-β expression in melanoma cells. We also observed increased expression of mesenchymal markers such as α-SMA, β-catenin, and vimentin in Treg-co-cultured B16-BL6 derived tumor tissues (Figure 4B,C), in agreement with previous reports demonstrating that increased α-SMA expression in cancer cells leads to invasion and metastasis of many epithelial cancers [42,43]. Of note, Treg-co-cultured B16-BL6 derived tumor tissues showed nuclear translocation of β-catenin as well as rearrangement of vimentin into dense structures around the perinuclear region (Figure 4B,C). These results are indicative of EMT induction; previous reports have shown that EMT requires translocation of β-catenin into the nucleus [44,45,46,47] and perinuclear accumulation of vimentin [48,49,50,51]. Furthermore, the expression of the transcription factors, Snail and Slug, which are known to promote a mesenchymal phenotype [52], was also markedly increased in Treg co-cultured B16-BL6 derived tumor tissues. Thus, these results strongly suggest that Tregs may increase cancer metastasis through the upregulation of TGF-β and mesenchymal markers. 

### 3.5. Increased Migration of Cancer Cells from Either Established Treg Co-Cultured B16-Bl6 Derived Tumors or Treg-Injected Tumors

The migration of cancer cells dissociated from Treg-co-cultured B16-BL6-derived tumors, as well as from Treg-injected tumors, was used as an indicator to assess the EMT-induced metastatic potential of each group. As shown in Figure 5A, cancer cells from Treg-co-cultured B16-BL6 derived tumors showed greater migration capacity than were cancer cells from tumors derived from B16-BL6 cells cultured without Tregs (*p* < 0.001). This result shows that Treg co-cultured melanoma cells acquire metastatic potential that is retained *in vivo*. 

Next, we assessed whether the injection of exogenous Tregs, which stably express GFP, could increase the metastatic potential of melanoma. B16-BL6 tumor-bearing mice were injected intratumorally with Tregs (2 × 10^7^ cells) 3 times every other day. As shown in Figure 5B, cancer cells from Treg-injected tumors showed a greater ability to migrate than did cells from PBS-injected tumors (*p* < 0.01). In addition, a higher proportion of Tregs was observed in the population of tumor-infiltrating lymphocytes from mice injected with exogenous Tregs compared to mice treated with PBS (Figure 5C; *p* < 0.001). Of note, the exogenous Tregs were not detected in the tumor infiltrating lymphocyte population (data not shown), suggesting that the increase in the Treg population in the B16-BL6 + Treg group was caused by exogenous Tregs promoting the recruitment of endogenous Tregs to the tumor microenvironment. 

### 3.6. Promotion of Tumor Metastasis by Treg Injection

The effect of Tregs on cancer metastasis was evaluated using a spontaneous metastasis model. B16-BL6 cells were subcutaneously injected into the footpad to establish a primary tumor. Three injections of Tregs (1 × 10^6^ cells) were intratumorally administered to the primary tumor, which was then excised to accelerate metastasis of melanoma to the lung. As shown in Figure 6A, the metastatic tumor volume and the number of metastatic nodules in the lung was greater following injection with Tregs than with PBS. The metastatic burden in the lungs was indirectly analyzed by measuring the overall weight of the combined normal and cancerous tissues. This approach was utilized due to the difficulty of precisely excising metastatic nodules from adjacent normal lung tissues. As shown in Figure 6B, the average weight of lungs from Treg-injected mice (800.0 ± 72.1 mg) was 2.6-fold higher than the weight of those from PBS-injected mice (310.0 ± 79.4 mg) (*p* < 0.01). Taken together, these results suggest that intratumoral injection of Tregs at the primary tumor promotes the formation of metastatic lesions at distal sites, ultimately showing that Tregs are a key inducer of melanoma metastasis.

## 4. Discussion

Metastasis causes as much as 90% of cancer-associated mortality, yet this process remains one of the poorly understood components of cancer pathogenesis [53]. Despite progress in the early diagnosis of malignant melanoma, a substantial percentage of patients present with distant metastases [54]. Manifestation of distant metastases in melanoma patients is associated with a poor prognosis and this subset of patients has a median survival period of 5 to 8 months [55,56].

Thus, the molecular and cellular mechanisms of metastasis are being extensively investigated to better predict, identify, and eradicate disseminated cancer [57]. However, metastasis does not occur solely as a result of the innate attributes of the tumor cell; the tumor microenvironment is a key contributor to malignant transformation and dissemination of the disease [58,59,60]. Additionally, a multitude of different cell types (such as a subset of stromal cells and immune cells) also exacerbate tumor cell growth and invasion and ultimately induce metastasis [9]. In this study, we differentiated Tregs from cells derived from GFP transgenic mice and, through co-culture or intratumor injection, investigated the effects of these Tregs on the invasive/metastasizing phenotype of melanoma cells. 

We demonstrate that Tregs can expedite melamona metastasis through a mechanism that involves cell-to-cell contact; direct cell-to-cell interactions between Tregs and melanoma cells elevated the TGF-β expression level in melanoma cells and subsequently promoted their migration and invasion (Figure 1, Figure 2, Figure 4A, and Figure 5), whereas inhibition of contact prevented Treg-mediated induction of TGF-β expression in melanoma cells (Figure 2B). Direct cell-to-cell interactions between cancer cells and other cell types have previously been shown to be necessary for increased invasive activity by cancer cells. For example, direct contact between mesenchymal stem cells and human carcinoma cells promotes invasive and metastatic properties of cancer cells, emphasizing that contact-based signaling activation is critical for phenotypical changes in melanoma cells [61]. Furthermore, we observed that exogenous Tregs administered to B16-BL6 tumors increased the infiltration of endogenous Tregs into tumor tissues even when the exogenous Tregs no longer remained in the tumor (Figure 5C). These findings suggest that tumor-infiltrating Tregs may self-promote recruitment of other circulating Tregs in the tumor microenvironment by inducing TGF-β expression in melanoma cells.

The relationship between EMT and metastasis has been of immense importance in the field of cancer research, with numerous reports illustrating a strong positive correlation between the two factors [62,63]. Although the relationship has been clearly shown in various reports, only a limited number have investigated how Tregs affect EMT in melanoma cells. In the current study, we found that melanoma cells augmented the expression of various mesenchymal phenotypic markers, such as Smad2/3, α-SMA, and vimentin, in addition to that of TGF-β, Snail, and Slug, following exposure to Tregs (Figure 3A,B and Figure 4B,C). Moreover, we noted strong nuclear translocation of β-catenin and perinuclear condensation of vimentin, which is another hallmark of EMT, in the Treg-co-cultured B16-BL6 tumor tissues (Figure 4B,C). These findings are in line with recent work demonstrating that concomitant elevation in the expression level of both Snail and Slug promotes the migratory and invasion abilities of cancer cells and contributes to lymph node metastasis and poor cancer prognosis [64]. Collectively, these observations show that Tregs can increase the metastatic potential of melanoma through induction of a mesenchymal phenotype, which may depend on the paracrine factor TGF-β.

Notably, Tregs induced Foxp3 expression in melanoma cells with higher invasiveness and metastatic potential (Figure 1, Figure 3A, and Figure 6), suggesting that Foxp3 may play a pivotal role in Treg-induced metastasis. Indeed, the induction of Foxp3 expression by a subset of cancers in the clinic has also been shown to promote metastasis through increased proliferation, migration, and invasion [21,65,66,67,68,69]. For instance, overexpressed Foxp3 in non-small cell lung cancer can act as a co-activator to facilitate the Wnt/β-catenin signaling pathway, leading to the induction of EMT and the stimulation of tumor growth and metastasis [70]. In line with these findings, we further revealed that Treg-mediated induction of EMT in melanoma cells led to the increased expression of the protease MMP9 (Figure 3C), suggesting that these cells may migrate to distal sites by degrading various physical barriers deterring cell migration. There are strong lines of evidence to support this claim, because increased expression of extracellular matrix-degrading proteases, such as MMP9, is a prerequisite for the invasive phenotype of cancer cells [71,72]. These EMT-induced cells with mesenchymal phenotypes have been reported to exhibit aberrant adhesive properties, amplified secretion of extracellular proteases, and a transformed extracellular matrix protein expression profile [73]. 

In conclusion, our study provides further insight into the mechanisms involved in melanoma metastasis, revealing novel aspects of the reciprocal contacts between melanoma cells and Tregs. This is the first report to show that physical contact between melanoma cells and Tregs is integral to the elevation of TGF-β expression and the subsequent induction of EMT in melanoma cells, which induces further recruitment of Tregs nearby the tumors and enhances the overall migratory/invasive properties of melanoma cells. Collectively, our findings illustrate that Tregs induce TGF-β-mediated EMT and subsequent metastasis of melanoma cells, suggesting that Tregs may serve as a promising therapeutic target and a predictive biomarker for melanoma metastasis.

## Figures and Tables

**Figure 1 cells-08-01387-f001:**
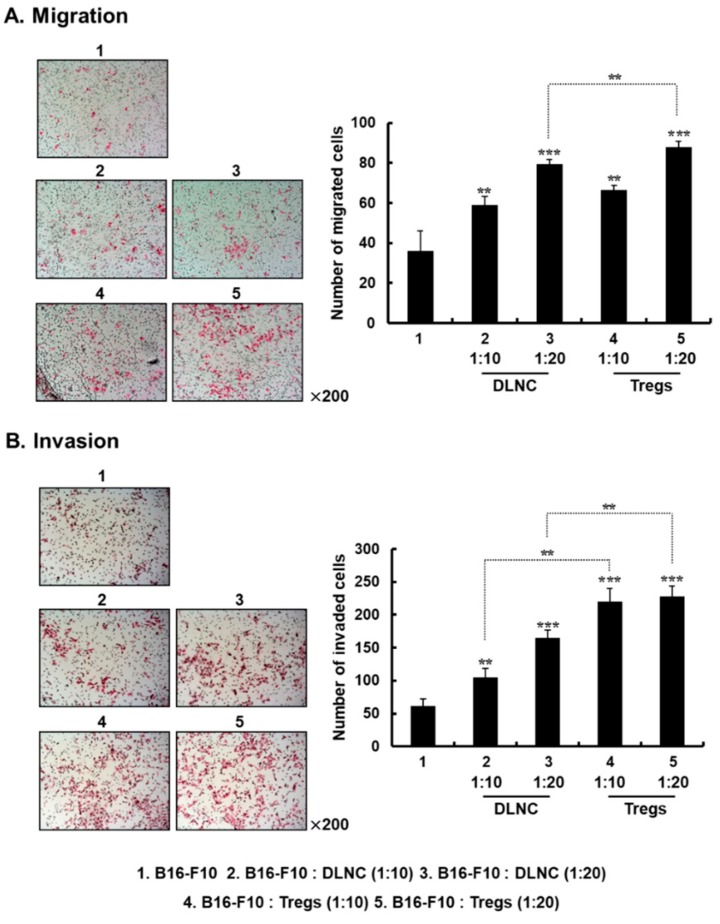
Effect of regulatory T cells (Tregs) on the migration and invasion of B16-F10 cells. Cell migration (**A**) and invasion (**B**) assays were conducted using a modified transwell migration chamber. After B16-F10 cells were co-cultured with draining lymph node cells (DLNC) or Tregs at various co-culture ratios, cancer cells were placed on the filter membrane in the top chamber. Cells that had migrated or invaded to the lower filter surface were stained with hematoxylin and eosin (H & E). Images are representative of results from three independent experiments performed in triplicates. Original magnification: ×200. Cells that had migrated or invaded were counted in three randomly selected fields from three independent wells for each experimental group. Data are shown as mean ± standard deviation (SD) of results from three independent fields/well. Figure 1A: ** *p* < 0.01 or *** *p <* 0.001 versus B16-F10 group, and ** *p* < 0.01 for B16-F10: DLNC versus B16-F10: Tregs (1:20). Figure 1B: ** *p* < 0.01 or *** *p <* 0.001 versus B16-F10 group, and ** *p* < 0.01 for B16-F10: DLNC (1:10) versus B16-F10: Tregs (1:10) and B16-F10: DLNC (1:20) versus B16-F10: Tregs (1:20).

**Figure 2 cells-08-01387-f002:**
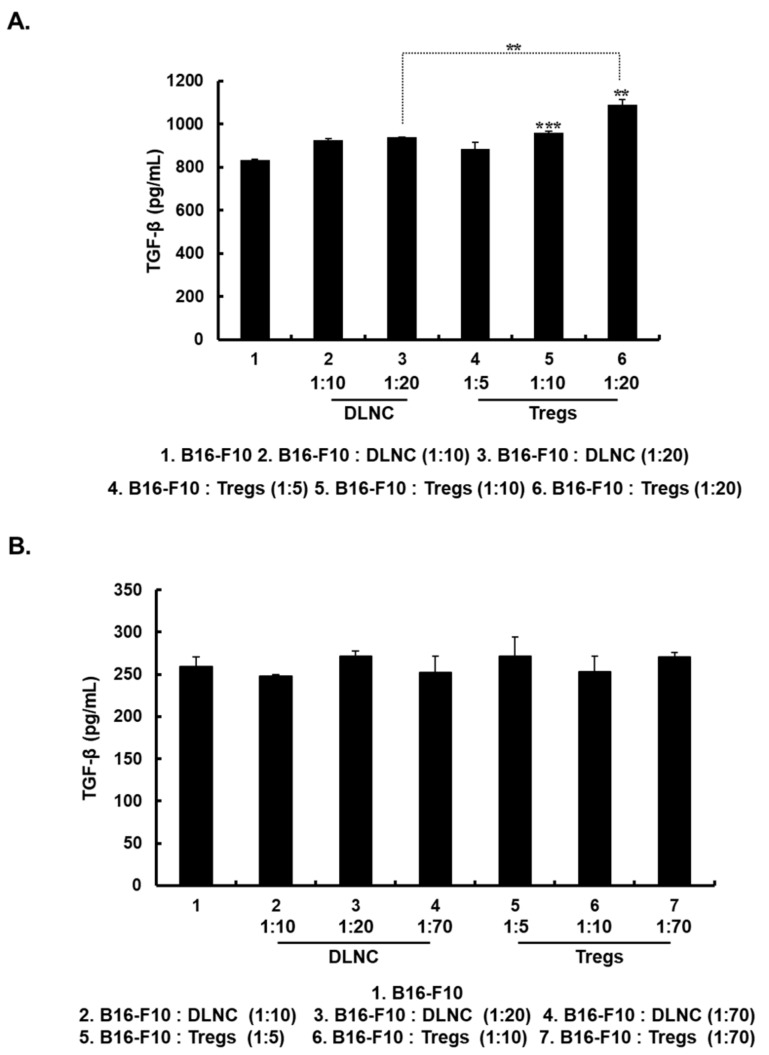
Expression of transforming growth factor-β (TGF-β) in B16-F10 cells. (**A**) B16-F10 cells were co-cultured with DLNC or Tregs at various co-culture ratios in 6-well plates and the TGF-β expression level in culture supernatants was measured at 72 h after incubation began by enzyme-linked immunosorbent assay (ELISA). Data are presented as mean ± SD of triplicates that are representative of 3 independent experiments. ** *p* < 0.01, B16-F10: Tregs (1:20) versus B16-F10 or B16-F10: DLNC (1:20). *** *p* < 0.001 versus B16-F10. (**B**) B16-F10 cells were physically separated from co-cultured DLNC or Tregs by 24-well transwell chamber. TGF-β expression level in the culture supernatant of lower chamber was measured at 72 h after incubation by ELISA. Data are presented as mean ± SD of triplicates that are representative of 3 independent experiments.

**Figure 3 cells-08-01387-f003:**
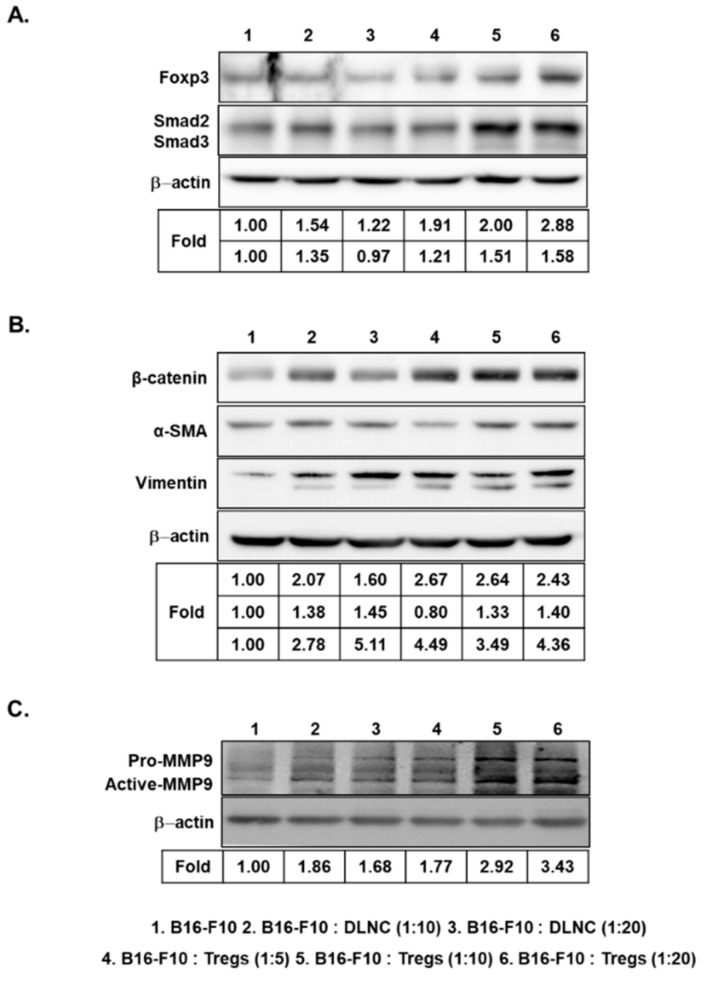
Increased expression of forkhead box P3 transcription factor (Foxp3), Smad2/3, epithelial–mesenchymal transition (EMT) markers, and matrix metalloproteinase 9 (MMP9) in B16-F10 cells following co-culture with Tregs. Representative Western blot analysis of Foxp3 and Smad2/3 (**A**), mesenchymal markers (β-catenin, alpha-smooth muscle actin (α-SMA), and vimentin) (**B**), and pro- and active-MMP9 (**C**) from the B16-F10 cell lysate harvested at 72 h after the initiation of co-culturing with DLNC or Tregs at various co-culture ratios.

**Figure 4 cells-08-01387-f004:**
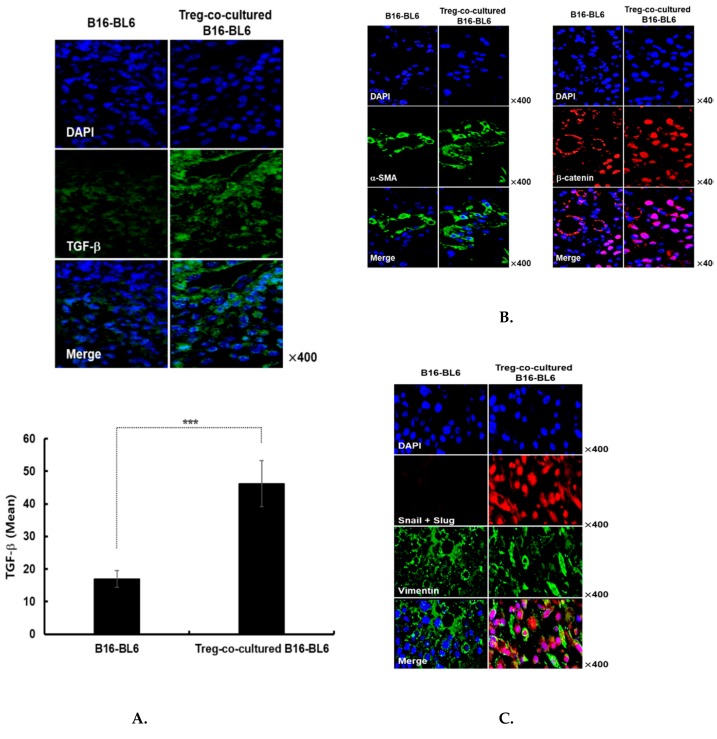
Immunohistochemical assessment of tumors resulting from subcutaneous injection of Treg co-cultured B16-BL6 cells. Tumor tissues were stained with anti- (**A**) TGF-β, (**B**) α-SMA and β-catenin, or (**C**) Snail, Slug, and vimentin antibodies. The TGF-β expression level in tissues was semi-quantitatively analysed by ImageJ software using 5 independent fields within microscope images for each experimental group. Data are presented as mean ± SD. *** *p* < 0.001. Original magnification: ×400.

**Figure 5 cells-08-01387-f005:**
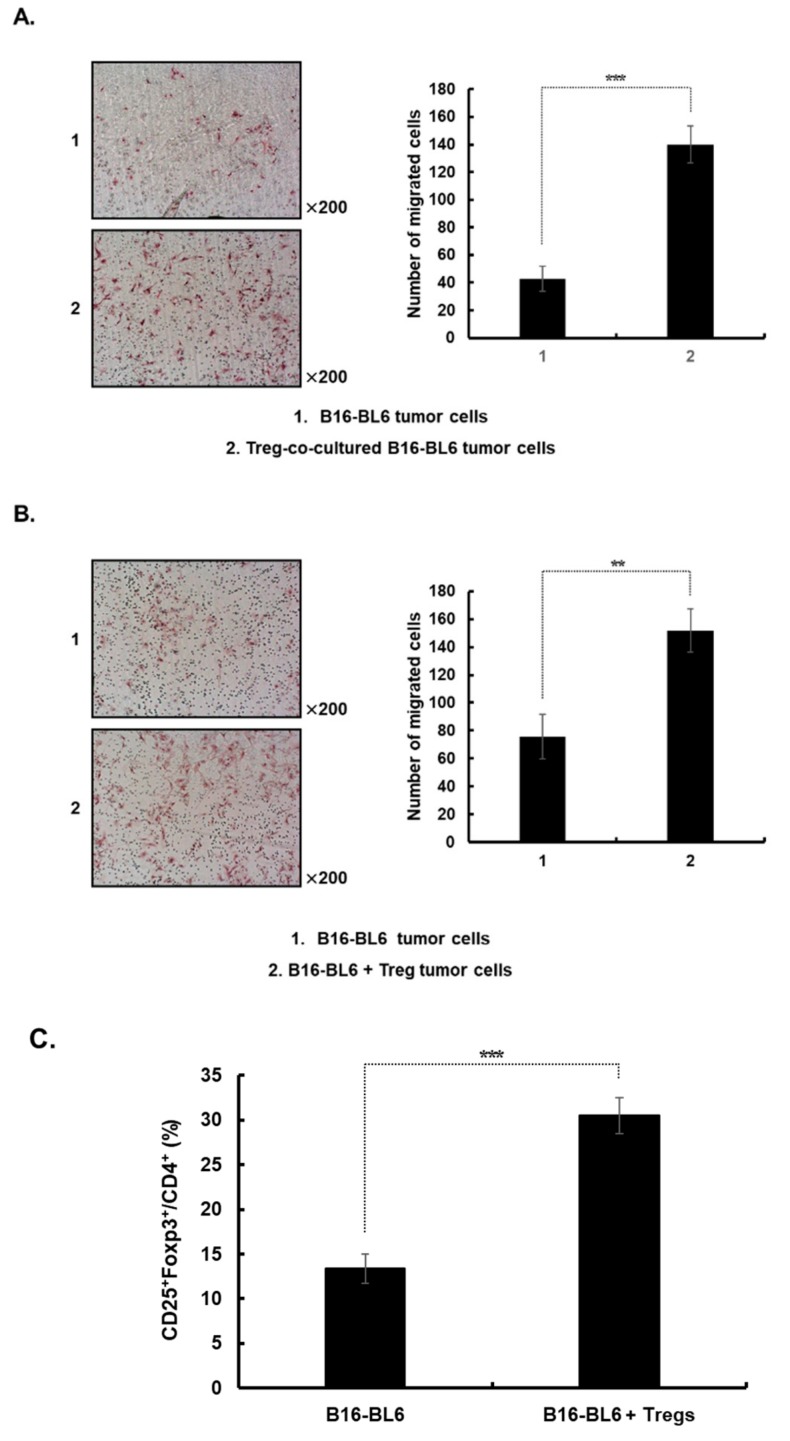
Increased migration of cancer cells and Tregs in both tumors established from Treg-co-cultured B16-BL6 cells or tumors injected with Tregs. To assess the effect of Tregs on the migration of dissociated tumor cells, (**A**) B16-BL6 cells were co-cultured with Tregs for 72 h at co-culture ratios of 1:10 when inadherent Tregs were removed from culture by multiple washing steps. Subsequently, the cells were detached and 5 × 10^5^ cells were injected subcutaneously into the right abdomen to establish a tumor or (**B**) subcutaneous tumors in the right abdomen were established using naïve B16-BL6 cells and subsequently injected intratumorally with Tregs (2 × 10^7^ cells) three times every other day. The migration profile of cells dissociated from the B16-BL6 tumors was analyzed using a modified transwell migration chamber. The cells that had migrated to the lower filter surface were stained with H & E. The images are representatives of results from 3 independent experiments. Original magnification: ×200. Migrated cells were counted in three randomly selected fields. Data are shown as mean ± SD of results from three independent fields/well. *** *p* < 0.001 or ** *p* < 0.01. (**C**) Tumors were collected from mice injected with Tregs at 5 days after the final injection. The populations of Tregs in tumors from mice were analyzed by flow cytometry. Gating was for cluster of differentiation (CD)4^+^ T cells and analysis for CD25^+^ and Foxp3^+^ cells. The data are representative of three independent experiments performed in triplicate. Data points are mean ± SD (*n* = 3 per group). *** *p* < 0.001.

**Figure 6 cells-08-01387-f006:**
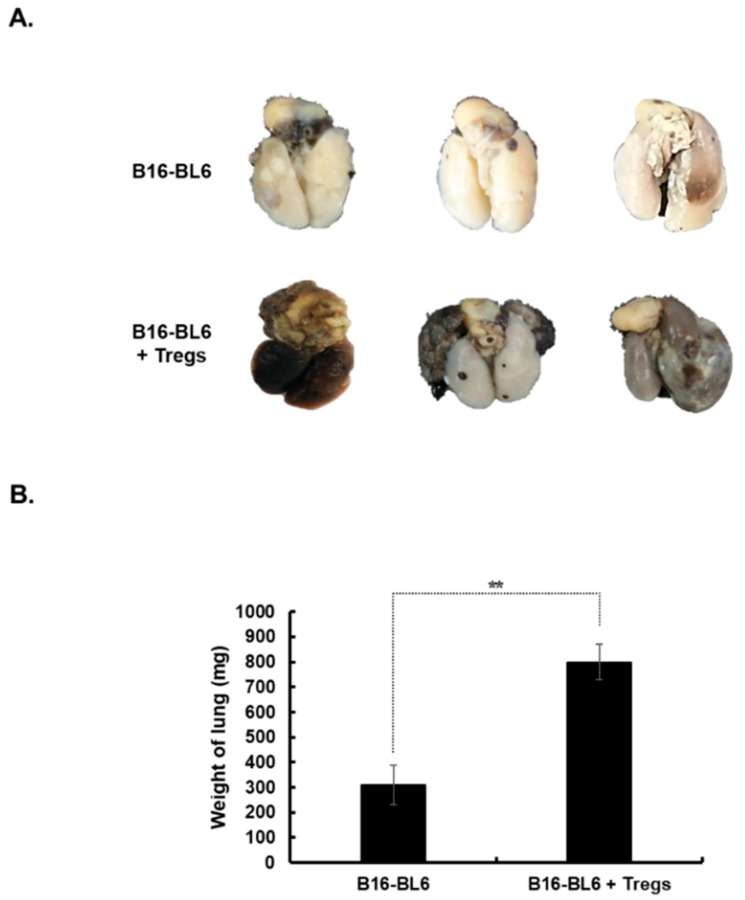
Induction of metastasis following injection with Tregs. Subcutaneously established B16-BL6 tumors located on the right foot pad were intratumorally injected three times with Tregs (1 × 10^6^ cells) and then surgically excised at 5 days after the final Treg injection. On day 25 after primary tumor removal, the lung tissues encompassing both normal and cancerous regions were weighed. (**A**) Representative macroscopic view of lungs from each group. These images are representatives from three independent experiments. (**B**) Comparison of the lung tissue weight from each group. Data represent mean ± SD of results from three mice per group. ** *p* < 0.01.

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
