# Peer review of "Regulatory T Cells Induce Metastasis by Increasing Tgf-β and Enhancing the Epithelial–Mesenchymal Transition"

_cells, 2019, doi:10.3390/cells8111387_

Round 1

Reviewer 1 Report

This is a very interesting report, and for the most part the data are convincing, although they are limited to a single B16 cell model. There are methods and results that are not clearly described, however and more details are required in the methods and in the description of the results - I have itemised these below:

It is not clear from where the DLNC are derived, and what these cells are? Can the authors provide some details on these cells - if they are simply cells derived from DLN of mouse melanoma models, then some FLOW analysis to indicate the types of cells present would be important.

The invasion and migration assays shown Figure 1 are not described in sufficient detail, i.e. I am not sure what the difference between the two assays are  in this Figure. The statistics used in the comparisons should be stated in each figure legend (throughout manuscript) to improve clarity. It is also unclear how many times the Figure 1 experiments were conducted - so 3 fields per well - how many wells per experiment and how many experiments. Why was t test chosen here for multiple  comparisons?

I am curious as to why the TGFß values are so different between Fig 2A and 2B, compare samples #1 for each - does this suggest that there could be methodological reasons for the low TGFß levels in 2B? rather than the conclusion that direct contact of cells is necessary for TGFß increase. If the Figure 2A experiment was conducted using transwell, with cells cocultured in the top well, would the lower chamber media show increased TGFß expression?

The data in Figure 3 needs to be quantitated - if conclusions are that markers are increased by co-culturing. and details of the antibodies used should also be provided - clone numbers. I am a little surprised that SMAD 3 increases in presence of co-culture - unless this is p-SMAD3 being investigated in these westerns? The MMP9 data is not convincing and again if conclusions regarding expression are to be made the authors should show protein quantitation.

In section 3.5 the authors should clarify the two types of melanoma models used. The TReg injected tumours are described in the methods, but the TReg-co-culture B16 derived tumours are not described - how where they generated, i.e. culturing details and injection/cell number information

The methods state TRegs injected at 1x106 cells, and in section 3.5 states 2x107 cells 3 times every other day - the method details are not clear throughout and need to be reviewed. Also, there is no mention of introducing GFP into TRegs in the methods- I assume they were derived (how) from the GFP-positive mice? This needs to be clarified.

Not sure what Table 16 refers to in Figure 5

Reviewer 2 Report

I have read a manuscript by Eonju Oh, Jin Woo Hong and Chae-Ok Yun entitled “Regulatory T cells induce metastasis by activating TGF-β and enhancing the epithelial-mesenchymal transition”, submitted to the Cells journal.

This study is the first showing the co-culturing of melanoma cells and Tregs results in the increase in TGF-β expression (as well as Snail and Slug) and EMT (Smad2/3, α-SMA, and vimentin ) in melanoma cells.

This finding is very interesting and having a lot of clinical implications. The competence of Treg cell has been recently investigated and published d in the paper by Ahmadzadeh M et al. Sci Immunol. 2019 Jan 11;4(31), which means that repeirtoire of tumor infiltration CD4 positive lymphocytes.

This manuscript is well written and has a good merit. However, some methodologic approach is not clear and needs more explanation.

Please, elucidate and justify the applied methodology for Tregs isolation, purification, and differentiation from draining lymph nodes (2.3). Why the authors did not choose the more selective method of cell sorting using a cell sorter or magnetic beads? It is very important to characterize those exogenous Treg cell with respect their surface antigens (CD25) which expression should be confirmed after IL-2 stimulations. Do the authors have any flow cytometry data presenting antigens expression of Treg cells before applying them in a spontaneous lung metastasis model? If not, such analyses are necessary to be performed. Western blot study has been performed on B16-F10 cell exposed to Tregs in different proportions. There is no in vivo model of this transition. Can the authors suggest such and experiment? Did the authors observed an increase in circulating TGF-β in mice?

Minor comment

Please include catalog numbers of antibodies used in staining protocols

Round 2

Reviewer 2 Report

I found a manuscript much improved